# Impact of Clinical Response to Neoadjuvant Chemotherapy in the Era of Robot Assisted Radical Cystectomy: Results of a Single-Center Experience

**DOI:** 10.3390/jcm9092736

**Published:** 2020-08-24

**Authors:** Umberto Anceschi, Aldo Brassetti, Gabriele Tuderti, Maria Consiglia Ferriero, Manuela Costantini, Alfredo Maria Bove, Fabio Calabrò, Paolo Carlini, Sabrina Vari, Riccardo Mastroianni, Michele Gallucci, Giuseppe Simone

**Affiliations:** 1Department of Urology, Regina Elena National Cancer Institute, Via Elio Chianesi 53, 00144 Rome, Italy; aldo.brassetti@gmail.com (A.B.); gabriele.tuderti@gmail.com (G.T.); marilia.ferriero@gmail.com (M.C.F.); m.costantini@unicampus.it (M.C.); alfredo.bove@yahoo.it (A.M.B.); puldet@gmail.com (G.S.); 2Department of Oncology, San Camillo-Forlanini Hospital, Circonvallazione Gianicolense 87, 00152 Rome, Italy; fabiocalabro1@alice.it; 3Department of Oncology, Regina Elena National Cancer Institute, Via Elio Chianesi 53, 00144 Rome, Italy; paolo.carlini@ifo.gov.it (P.C.); sabrina.vari@ifo.gov.it (S.V.); 4Department of Urology, La Sapienza—University of Rome, Viale del Policlinico 155, 00161 Rome, Italy; riccardomastroianniroma@gmail.com (R.M.); michele.gallucci@uniroma1.it (M.G.)

**Keywords:** neoadjuvant chemotherapy, clinical response, robotic radical cystectomy, muscle-invasive bladder cancer, lymphadenectomy, overall survival, urothelial carcinoma

## Abstract

Background: Response to neoadjuvant chemotherapy (NACT) has been proven to be an established prognostic factor after open radical cystectomy (ORC). We evaluated the impact of NACT on survival outcomes of a single-institution robotic radical cystectomy (RARC) series. Methods: From January 2012 to June 2020, 79 patients were identified. Baseline, demographic, perioperative, and pathologic data were described. Kaplan–Meier with the log-rank test was used to compare overall survival (OS) differences between complete, partial, and no-NACT responders, respectively. Univariable and multivariable regression analyses were performed to identify predictors of OS. Results: Complete, partial, and absent response to NACT were recorded in 43 (54.4%), 21 (19%), and 15 (26.6%) patients, respectively. A complete response to NACT displayed a trend toward significant higher OS (*p* = 0.03). In univariable analysis, significant predictors of lower OS were hypertension (HR 3.37; CI 95% 1.31–8.62; *p* = 0.01); advanced nodal involvement (HR 2.41; CI 95% 0.53–10.9; *p* < 0.001); and incomplete response to NACT (HR 0.41; CI 95% 0.18–0.95; *p* = 0.039). In multivariable analysis, the only independent predictor of worse OS was advanced pathologic N stages (HR 10.1; CI: 95% CI 2.3–44.3; *p* = 0.002). Conclusions: Complete response to NACT is associated with increased OS probability, but significant nodal residual disease remains the only independent predictor of OS after RARC.

## 1. Introduction

Radical cystectomy (RC) with pelvic lymph node dissection represents the standard of care for patients with muscle-invasive urothelial bladder cancer (MIBC), recurrent, or Bacillus Calmette-Guerin (BCG)-refractory non-muscle-invasive high-grade UC [1]. With increasing adoption of advanced and minimally-invasive technologies, robot-assisted RC (RARC) has gained wider consensus among tertiary-care centers, achieving comparable oncological and functional outcomes to conventional open surgery [1,2,3]. Nowadays, for patients with MIBC, a multidisciplinary approach including neoadjuvant chemotherapy (NACT) significantly enhances the probability of cure [4]. According to the most recent European Association of Urology (EAU) and the National Comprehensive Cancer Network guidelines (NCCN), administration of NACT improves overall survival (level 1 evidence) and should be considered in those patients who are most likely to develop recurrent disease [5,6]. Despite the fact that NACT should represent a cornerstone of MIBC, guideline-driven neoadjuvant regimens are administered in approximately only in 16–26% of cases [7]. Reasons for underutilization of NACT are usually represented by host clinical factors as increased toxicity over multiple comorbidities, advancing age, or poor performance status [8]. Nonetheless, concerns regarding associated peri-operative morbidity and surgical treatment delay in non-responder patients may further represent additional reasons for a suboptimal utilization of NACT [9]. Since response to guideline-driven neoadjuvant regimens has been proven to be an established prognostic factor after open radical cystectomy (ORC), data reported on this issue in the era of robotic radical cystectomy (RARC) are scanty and limited to multicentric series [10,11]. In this context, we sought to evaluate the impact of NACT on perioperative and survival outcomes of RARC at a single high-volume institution. The secondary endpoint of the study was to identify predictors of overall survival (OS) after RARC.

## 2. Materials and Methods

Our prospectively-maintained RARC dataset was queried for, “neoadjuvant chemotherapy (NACT)”and “totally intracorporeal urinary diversion”. From January 2012 to June 2020, out of a total of 254 RARC, 79 patients matching the inclusion criteria were identified. Our surgical techniques for RARC with totally intracorporeal urinary diversion were previously described elsewhere [12,13,14]. Patients receiving suboptimal NACT regimen (<3 cycles) due to toxicity (or any other reason for discontinuation) and patients with any mixed or variant histology of pure urothelial carcinoma (UC) were excluded from the study. Prior radiotherapy for MIBC, palliative cystectomy, concomitant nephroureterectomy, or urethrectomy as missing data were also considered as exclusion criteria. Indication to surgery was elective in all cases. All patients were preoperatively evaluated with a total-body computed tomography scan (CT) before surgery. Patients were categorized according to NACT using the Response Evaluation Criteria in Solid Tumors V 1.1 (RECIST V.1.1) guidelines in three groups: complete responders (those who achieved after NACT a complete pathological response as ypT0N0 on final pathology; *n* = 43), partial responders (patients who showed after NACT a downsizing of disease at preoperative imaging or achieving a significant downstaging of disease on final pathology; *n* = 15), and the non-responders group (patients who showed after NACT no downstaging of disease at preoperative imaging or not achieving a significant downstaging of disease on final pathology; *n* = 21) [15].

Baseline, demographic, perioperative, pathologic, functional, and oncologic data were included in the analysis. Evaluated preoperative clinical and demographic characteristics included age, gender, body mass index (BMI), American Society of Anesthesiologists (ASA) score, smoking status, medical comorbidities (diabetes, hypertension), cT stage, baseline estimated glomerular filtration rate (eGFR ml/min/1.73 m2), baseline chronic kidney disease (CKD), type of urinary diversion (continent/incontinent), and median duration from NACT completion to RARC [16,17]. Perioperative variables included mean hemoglobin drop (ΔHb), % perioperative complications, according to the Clavien Grading System [18]. Pathological outcomes were considered pT stage, pN stages, lymph node yield, concomitant carcinoma in situ (CIS), and surgical margin status (PSM). Functional outcomes consisted of newly onset of CKD stage 3b, 4, 5 recorded at last follow-up. Oncologic outcomes were represented by overall survival (OS) and disease-free survival (DFS).

Primary endpoints of the study were to compare baseline, perioperative, pathologic outcomes, and OS between groups. Descriptive analyses were used. Frequencies and proportions were reported for categorical variables while medians and interquartile ranges (IQR) were reported for continuously coded variables. Differences between continuous variables were assessed using an unpaired t-test and one-way analysis of variance (ANOVA), while the Pearson χ^2^ test was used for categorical data. OS probabilities were computed by Kaplan–Meier curves and compared for NACT response with the log-rank test.

The secondary endpoint was to identify predictors of OS after RARC using univariable and multivariable Cox regression analysis. For all analyses, a two-sided *p* < 0.05 was considered significant. Statistical analysis was carried out using the Statistical Package for Social Sciences (SPSS) software v26.0 (IBM Corp, Armonk, NY, USA).

## 3. Results

Demographic and preoperative data are shown in Table 1.

Forty-three patients (54.4%) achieved a complete NACT response, 15 patients (18.9%) were partial responders, while NACT failure was reported in 21 patients (26.5%), respectively. No significant differences were found between groups in terms of age, gender, smoking status, medical comorbidities, BMI, ASA score, cT stage before NACT administration, preoperative hydronephrosis, type of urinary diversion, and median duration from completion of NACT to RARC (each *p* < 0.8). Partial responders showed a significant eGFR impairment (*p* = 0.03) at baseline with four patients (26.7%) reporting a preoperative CKD stage ≥ 3a.

Perioperative and pathologic outcomes are summarized in Table 2.

The overall complications rate and ΔHb were comparable between groups (each *p* < 0.6). According to the Clavien Grade distribution, the number of major complications did not reach significance between groups, ranging between 7–14.3% (*p* = 0.634). All complete responders to NACT achieved a complete pathological response (ypT0N0) with negative surgical margins and no concomitant Cis. In the partial responders series, nine patients had pTa-pT1 disease (60%) while pT2a-pT2b stages were reported in six patients (40%). In contrast, in the NACT failure group, 16 patients (76.2%) showed a locally-advanced or higher disease (*p* = 0.001). In both partial and no responders groups, patients had significantly higher rates of positive lymph nodes (pN+: 53% and 42.8%, respectively). There was no difference between groups in terms of lymph node yield and surgical margin status. Concomitant Cis was significantly higher in partial responders (*p* = 0.01).

At a median follow-up of 33 months (IQR 13–50.2), no difference was found between groups in terms of eGFR and CKD stage distribution (each *p* < 0.3). In the complete responders group, a disease recurrence was observed in 10 patients (23.6%), while nine patients (20.9%) died during the observation period (median 42 months IQR: 22.5–53 months). In the partial responders group, five patients experienced recurrent disease (33.3%) while six patients (40%) died at a median follow-up of 33 months (IQR: 13–44.5). Between no-responders, a disease recurrence was reported in nine patients (42.9%), while nine patients died at a median follow-up of 13.5 months (IQR: 5.2–49.5).

In the Kaplan–Meier analysis, patients who achieved a complete response to NACT displayed a trend toward significant higher OS probabilities compared to partial/no responders to NACT (Figure 1; *p* = 0.03); (Figure 1).

In the univariable analysis, hypertension (HR 3.37; CI 95% 1.31–8.62; *p* = 0.01), pathologic nodal stages N2–N3 (HR 2.41; CI 95% 0.53–10.9; *p* < 0.001), and incomplete response to NACT (HR 0.41; CI 95% 0.18–0.95; *p* = 0.039) were all significant predictors of worse OS. In the multivariable analysis, the only independent predictor of worse OS was residual pathologic N2–3 stages (HR 10.1; 95% CI 2.3–44.3; *p* = 0.002) (Table 3).

## 4. Discussion

Since it has been established that NACT before radical cystectomy improves survival for MIBC compared to RC alone, the best OS benefit was achieved by those patients with a complete pathologic response after NACT (ypT0N0), ranging between 20–38% of cases [19]. However, as the majority of patients experience a partial clinical downstaging or a negative response at the time of surgery after NACT, the prognostic value of residual MIBC disease (rMIBC) after RC still remains unclear [20,21]. Moreover, despite a robust literature on safety outcomes of RARC is already available, a prospective reporting of oncological outcomes after chemotherapy plus RARC is still limited to a few multicentric series and mainly focused on NACT dosage optimization and its impact on perioperative outcomes [11,22].

In this scenario, we examined 79 patients with pure MIBC of the bladder who underwent RARC with total intracorporeal diversion following NACT in order to identify predictors of OS according to different NACT pathologic response. To the best of our knowledge, this is the first study examining the prognostic role of rMIBC after NACT in patients undergoing RARC at a single high-volume center.

Our study showed interesting findings. The rate of major perioperative complications after RARC (Clavien III–V) was comparable between groups ranging between 7–14.3% (*p* = 0.634). This is in line with recent multicentric studies, describing a negligible impact of NACT on the perioperative surgical morbidity of RARC [23]. The low incidence of overall complications reported may be reflective of both yearly high RARC caseload of the center and the experience of the surgical team gained over years [24]. In our series, the rate of patients achieving a complete NACT response (54%) prior to RARC significantly differed from results of previous other studies where a complete pathologic response was observed in 20–38% of cases, despite observing comparable survival outcomes (79.1% vs. 64%) [25,26]. On the other hand, the number of patients with pN+ after RARC was comparable between the partial and no-responders groups (53.3% vs. 42.9%, respectively). This is in contrast with recent findings where the rates of occult nodal metastases in patients with downstaged no MIBC after NACT was approximately 5.4% [27]. In the absence of clinical predictors of clinical involvement, our data suggest that a large number of patients who would have otherwise been considered partial NACT responders still have a significant clinically undetected but pathologically proven burden of disease and may benefit an extended lymphadenectomy.

There are several potential explanations and implications for our data. Independent of the surgical approach considered, rMIBC after NACT may represent a clinical marker of highly aggressive disease and its presence at final specimen may suggest the presence of chemoresistant micrometastatic disease. Furthermore, the overall high median number of lymph node yield observed across groups (30; IQR 22–36) may be related to the intervariability of pathologist assessment or to an intrinsic bias of robotic approach. Furthermore, due to the heterogeneous design of our study, it is possible that several patients receiving NACT and RARC had inherently worse disease at diagnosis due to preoperative clinical factors that we were unable to retrieve in our dataset.

Despite all this, in multivariate analysis, after adjusting for major comorbidities and preoperative renal function, only advanced residual pN+ stages were independent predictors of worse survival prognosis (*p* = 0.002). Since in our series patients who achieved a partial response following NACT and non-responders showed comparable survival outcomes, these data suggest that the amount of rMIBC rather than the degree of downstaging following NACT is associated with subsequent survival outcomes. In analogy to previous ORC series, our study confirmed that patients with nodal metastatic disease after RARC have the same, worse, oncologic outcomes [27]. Thus, a critical evaluation of the response to NACT in patients with MIBC may be accurately predict the prognosis of the patients, or potentially the appropriate extent of the lymphadenectomy template.

Undoubtedly, several limitations warrant discussion. The non-randomized retrospective study design and the small number of patients with relatively short follow-up represent major drawbacks. Furthermore, due to the low number of patients included, we were unable to perform propensity-score-based matching in order to balance any confounders or systematic biases amongst groups that may play a role in explaining several perioperative differences or associations observed, as for hypertension. Additionally, although we did control our regression model for the main comorbidities, we were unable to provide any standardized frailty index or performance status in the multivariable analysis. Finally, the lack of comparison with pathologic stage-matched controls of patients who received alternative treatments (NACT with bladder-sparing strategy or RARC plus adjuvant chemotherapy or RARC alone) represent another potential source of bias.

Notwithstanding these limitations, we were able to demonstrate a direct and significant impact of different NACT response on survival, adding plausibility to a real OS benefit of NACT before surgery only when a complete pathologic response is achieved. Independently of MIBC chemosensitivity, the use of NACT did not seem to significantly impact perioperative morbidity of RARC in our series. Moreover, since residual lymph node disease (N2–N3) after NACT in our model represented the only independent predictor of worse prognosis following surgical treatment, these data reinforce the need for an extended lymph node template during RARC as a marker of surgical quality, which may provide a significant improvement in survival [28].

## 5. Conclusions

According to our experience, patients achieving a complete pathologic response to NACT showed increased OS probability after RARC, but significant nodal residual disease (pathological N2-3 stages) remains the only independent predictor of OS after RARC. The implementation of clinical tests to identify which patients may benefit from a complete NACT response as the clinical assessment of residual disease after NACT failure still remains an unmet need in the multimodal treatment of MIBC.

## Figures and Tables

**Figure 1 jcm-09-02736-f001:**
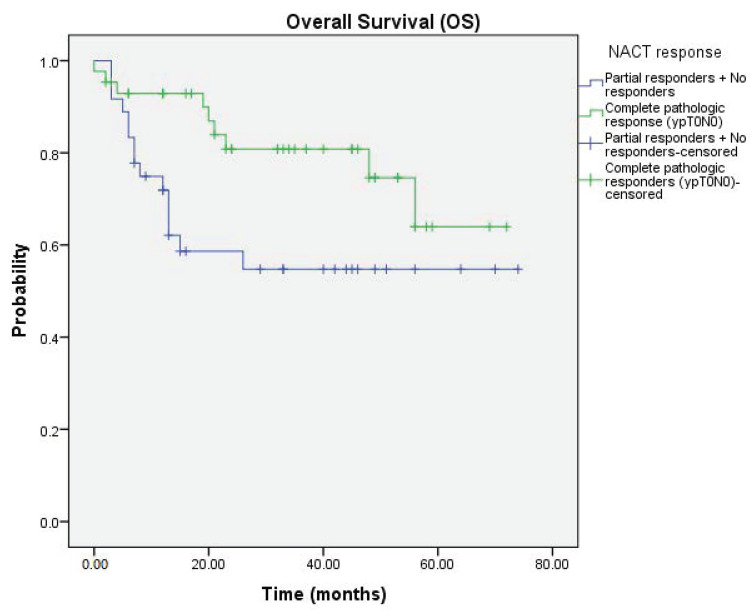
Kaplan–Meier curve showing OS probability according to NACT response.

**Table 1 jcm-09-02736-t001:** Patient demographic, clinical, and preoperative data according to NACT response ^1.^

Variable	Complete Responders (*n* = 43)	Partial Responders (*n* = 15)	No Responders (*n* = 21)	*p*
**Age (** **yrs, median, IQR)**	61 (56–66)	60 (52–67)	63 (54–66.5)	0.94
**Male gender (*n*, %)**	29 (67.4%)	9 (60%)	18 (85.7%)	0.193
**Smoker status (*n*, %)**	18 (41.9%)	5 (33.3%)	13 (61.9%)	0.289
**Diabetes (*n*, %)**	2 (4.7%)	3 (20%)	3 (14.3%)	0.117
**Hypertension (*n*, %)**	14 (32.6%)	7 (46.7%)	9 (42.9%)	0.293
**BMI (kg/m^2^, median, IQR)**	25 (23.5–27.3)	26 (20–29)	28.1 ± 5.6	0.619
**ASA score (*n*, %)**				
**1–2**	34 (79.1%)	13 (86.7%)	17 (81%)	0.818
**3–4**	9 (20.9%)	2 (13.3%)	4(19%)	
**Preoperative Hb (g/dL, median, IQR)**	12.9 (11.5–13.6)	12.8 (11.8–14)	13.7 (11.7–14.3)	0.704
**Preoperative eGFR (mL/min/1.73m^2^, median, IQR)**	78.5 (63.5–92)	64.7 (48.1–77.2)	85.1 (74.4–98.7)	0.03
**Preoperative hydronephrosis (*n*, %)**				
**No**	37 (86%)	11 (73.3%)	15 (71.4%)	0.501
**Left**	1 (2.3%)	1 (6.7%)	2 (9.5%)	
**Right**	4 (9.3%)	1 (6.7%)	1 (4.8%)	
**Bilateral**	1 (2.3%)	2 (13.3%)	3 (14.3%)	
**Preoperative nephrostomy (*n*, %)**	4	3	3	0.743
**Baseline CKD (*n*, %)**				
**1**	10(23.3%)	2 (13.3%)	9 (42.9%)	0.05
**2**	26 (60.5%)	9 (60.0%)	12 (57.1%)	
**3a**	7 (16.3%)	1 (6.7%)	-	
**3b**	-	2 (13.4%)	-	
**4**	-	1 (6.6%)	-	
**cT stage (*n*, %)**				
**2**	30 (69.8%)	9 (60%)	17 (81%)	0.651
**3**	12 (27.9%)	6 (40%)	3 (14.3%)	
**4**	1 (2.3%)	-	1 (4.8%)	
**Type of urinary diversion (*n*, %)**				
**Padua Ileal neobladder**	37 (86%)	11 (73.3%)	16 (76.2%)	0.729
**Indiana Pouch**	1 (2.3%)	2 (13.3%)	1 (4.8%)	
**Ileal conduit**	5 (11.6%)	2 (13.3%)	4 (19%)	
**Median duration from completion of NACT to RARC (days, *n*, IQR)**	35 (31–39)	34 (30–37)	37 (32–40)	0.654

^1^ Data are reported as median (IQR).

**Table 2 jcm-09-02736-t002:** Perioperative, pathologic, and functional data according to NACT response ^2^.

Variable	Complete Responders (*n* = 43)	Partial Responders (*n* = 15)	No Responders (*n* = 21)	*p*
**Overall complications (*n*, %)**	9 (20.9%)	4 (26.7%)	8 (38.1%)	0.354
**Clavien Grade (*n*, %)**				
**1–2**	7 (16.3%)	4 (26.7%)	7 (33.3%)	0.305
**3–5**	3 (7%)	4 (26.7%)	3 (14.3%)	0.634
**Median lymph node yield (*n*, IQR)**	27.5 (21–36.5)	35 (23–37)	31 (24–36)	0.859
**Positive surgical margins (*n*, %)**	-	3 (20%)	2 (9.5%)	0.028
**pT stage at cystectomy**		-		
**pT0**	43 (100%)	-	-	0.001
**pTa, pT1**	-	9 (60%)	-	
**pT2a**	-	2 (13.4%)	2 (9.5%)	
**pT2b**	-	4 (26.7%)	3 (14.3%)	
**pT3a**	-	-	3 (14.3%)	
**pT3b**	-	-	10 (47.6%)	
**pT4**	-	-	3 (14.3%)	
**pN**				
**N0**	43 (100%)	7 (46.7%)	12 (57.1%)	0.001
**N1**	-	3 (20%)	1 (4.8%)	
**N2**	-	4 (26.7%)	5 (23.8%)	
**N3**	-	1 (6.7%)	3 (14.3%)	
**Concomitant Cis (*n*, %)**	-	5 (33.3%)	3 (14.3%)	0.01
**Median Follow-up (months, median, IQR)**	42 (22.5–53)	33 (13–44.5)	13.5 (5.2–49.5)	0.125
**Last control eGFR (mL/min/1.73m^2^, median, IQR)**	62.6 (44.8–85.2)	59.7 (40–71.6)	64.4 (34.4–82.9)	0.292
**Last control CKD (*n*, %)**				
**1**	7 (16.3%)	1 (6.7%)	1 (4.8%)	0.253
**2**	18 (41.9%)	6 (40%)	11 (52.4%)	
**3a**	7 (16.3%)	2 (13.3%)	3 (14.3%)	
**3b**	11 (25.6%)	4 (26.7%)	5 (23.8%)	
**4**	-	1 (6.7%)	1 (4.8%)	
**5**	-	1 (6.7%)	-	
**∆Hb (median, IQR)**	1.5 (0.6–2.6)	1.1 (0.5–2.3)	2.2 (0.8–2.8)	0.598

^2^ Data are reported as median (IQR).

**Table 3 jcm-09-02736-t003:** Univariable and multivariable Cox regression analysis to identify predictors of OS.

Variable	Univariable Analysis				Multivariable Analysis (MIC)			
	HR	95% CI			HR	95% CI		
		Lower	Higher	*p* Value		Lower	Higher	*p* Value
**Age**	1	0.95	1.05	0.768	-	-	-	-
**Gender**	0.59	0.22	1.59	0.303	-	-	-	-
**ASA score** **1–2** **3–4**	1.74	0.68	4.43	0.239	-	-	-	-
**Smoking status**	0.98	0.39	2.4	0.966	-	-	-	-
**Diabetes**	2.49	0.90	6.87	0.07	-	-	-	-
**Hypertension**	3.37	1.31	8.62	0.01	2.23	0.83	6.04	0.111
**Preoperative Hydronephrosis**	2.39	0.80	7.11	0.117	-	-	-	-
**Type of urinary diversion (Continent vs Incontinent)**	2.20	0.81	5.96	0.121	-	-	-	-
**cT stage**	1.09	0.45	2.68	0.837	-	-	-	-
**Preoperative eGFR**	1	0.98	1.02	0.533	-	-	-	-
**pN stage (N1 vs. N2–N3)**	2.41	0.53	10.9	<0.001	10.1	2.3	44.3	0.002
**Median duration from completion of NACT to RARC**	0.9	0.82	1.11	0.563	-	-	-	-
**NACT (complete response vs. partial/no response)**	0.41	0.18	0.95	0.039	1.46	0.34	6.19	0.606

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
