# Peer review of "Impact of Clinical Response to Neoadjuvant Chemotherapy in the Era of Robot Assisted Radical Cystectomy: Results of a Single-Center Experience"

_jcm, 2020, doi:10.3390/jcm9092736_

Round 1
Reviewer 1 Report
This study was reported the utility of NACT in patients with MIBC who underwent RARC. The reviewer would like to suggest some critiques to make this paper as follows.
Major
- In the Abstract, “log-rank=0.03” is correct? In addition, the authors should spell out HR and CI.
- RC is the standard treatment for muscle-invasive bladder cancer. Muscle-invasive urothelial carcinoma is wrong.
- The definitions of partial responder, full response, and no response are unclear. The authors should evaluate the response for NACT using RECIST guideline
- In the Result, the reviewer could not understand the meaning of “a full NACT response”. The authors should revise this point.
- The reviewer think that OS is relatively low in patients with MIBC who achieved CR after NACT. In addition, the association between hypertension and OS is unclear. The authors should clarify these points.
Author Response
In the abstract, log-rank=0.03 is correct? In addition the authors should spell out HR and CI
We would like to thank the reviewer for this punctual comment. “Log-rank” was replaced with “p” and the value reported was right. The acronym “HR” represents the “hazard ratio” as the “CI” represents the “95% confidence interval”. We apologize for the wide use of acronyms in order to keep the word count around 200 words, as requested by the journal policy. We sincerely hope you may understand our point of view.
RC is the standard treatment for muscle-invasive bladder cancer. Muscle-invasive urothelial carcinoma is wrong.
Thank you for this suggestion. We replaced the terminology of muscle-invasive urothelial carcinoma with muscle-invasive bladder cancer, as requested.
The definitions of partial respomder, full response, and no response are unclear. The authors should evaluate the response for NACT using RECIST guideline.
We sincerely thank the reviewer for this recommendation. We included a sentence in the material and methods section describing the methodology used to assess the response to chemotherapy (RECIST V.1.1). References were also updated, accordingly.
In the results, the reviewer could not understand the meaning of “full NACT response”. The authors should revise this point.
Thank you very much for this criticism. The terminology “full responder” was used as “complete responder” to identify all patients that after completing NACT showed a complete response to chemotherapy with “pT0 disease” on final pathology after RARC. In the material and methods section, line 75-81 were revised in order to clarify this point. The terminology “full responder” was removed in order to avoid confusion.
The reviewer think that OS is relatively low in patients with MIBC who achieved CR after NACT. In addition the association between hypertension and OS in unclear. The authors should clarify these points.
We really appreciated this comment. As stated in the discussion section, OS rates were comparable to other major series reporting outcomes of NACT + RC. Furthermore, a recent study published by Ploussard et al on JCM showed 1-, 2-, and 5-year OS rates of 66.8%, 34.6% and 16.3% after NACT+RC which appear comparable to our series. This manuscript was included in the references.
With regard to association between hypertension and OS, we believe that it may represent a result of the lack of balance between groups, due to the intrinsic retrospective design and the small series considered. For this reason, we revised the discussion section (line 204-205) including this aspect in the study limitations.
Reviewer 2 Report
The authors set out to present a series of 79 patients who have had NAC in the era of Robotic Cystectomy. The paper is well written and well presented.
However, the authors need to consider a revision addressing the following points
- Please define response accurately. I need to know the criteria being used for this. The protocol of scanning during cycles of NAC and post completion of NAC.
- Did the patients have any biomarker testing evolved during the June 2012- June 2020. Did the authors use Neutrophil : Lymphocyte ratios
- Please highlight the timing and correlation of the complication of patients and time of Robotic Cystectomy to completion of neoadjuvant chemotherapy. This needs to be discussed.
- The Baseline CKD rates are high in Table 1. Did any cohort of patients have stents / Nephrostomy pre-operatively ? If yes please present the number of patients who developed upper tract TCC. This will be interesting to look at as the series has long term data
- Please present data on stricture rates as some controversy exists on increase strictures Robotically.
- Please present any additional data on how many of the diversions are intra / extra corporeal as this is not very clear in the paper
Author Response
Please define response accurately. I need to know the criteria being used for this. The protocol of scanning during cycles of NAC and post completion of NAC.
Thank you very much for this suggestion. We believe to have already answered to a similar comment from reviewer 1. We included a sentence in the material and methods section describing the methodology used to assess the response to chemotherapy (RECIST V.1.1). As previously stated, references were also modified, accordingly.
Did the patients have any biomarker testing evolved during the June 2012-June 2020. Did the authors use Neutrophil/Lymphocyte ratios.
This is a very interesting point. Unfortunately, these data were missing in our dataset and consequently we were unable to test the predictive role of neutrophil/lymphocyte ratio in our series.
Please highlight the timing and correlation of the complication of patients and time of robotic cystectomy to completion of neoadjuvant chemotherapy. This needs to be discussed.
We would like to thank the reviewer for this criticisim. The median duration from completion of NACT to RARC was described in the text and it has been reported in tables, wherever appropriate. Furthermore, we tested this variable in the Cox regression analysis showing no significance (p=0.563) at univariable analysis.
The baseline CKD rates are high in Table 1. Did any cohort of patients have stents/nephrostomy preoperatively? If yes, please present the number of patients who developed upper tract TCC. This will be interesting to look at as the series has long-term data.
We appreciate this criticism. We reported in table 1 the number of patients who had a preoperative nephrostomy. Preoperatively, none of the patients underwent an indwelling ureteral catheter positioning. In the timeframe considered none of the patients developed upper tract TCC.
Please present data on stricture rates as some controversy exists on increase strictures robotically.
This is a good point. In the overall series considered, a single-side ureteral stricture was described in 5 cases (6.3%). We omitted to report this data in the manuscript since the objectives of the study were to evaluate the impact of NACT on perioperative outcomes (≤30 days) and OS. Ureteral strictures represent a common late complication of RARC (>90 days).
Please present any additional data on how many of the diversions are intra/extra corporeal as this is not very clear in the paper
Thank you for this precise comment. All patients underwent a totally intracorporeal urinary diversion in all cases. We underlined this aspect in the material and methods sections including detailed references. Thus, we feel that it would be unnecessary to provide any significant change in the material and methods section according to this aspect. Thank you.
Round 2
Reviewer 1 Report
None
Reviewer 2 Report
Accept